# Behavior test for seven-week old puppies (*Canis familiaris*): Inter-rater reliability and factors associated with test performance

**Daniela Alberghina**[1]*, **Fabiola Giunta**[1], **Mauro Gioè**[2,3], **Michele Panzera**[1]

**1** Department of Veterinary Science, University of Messina, Messina, Italy, **2** Department of Brain and Behavioral Sciences, University of Pavia, Pavia, Italy, **3** Department of Global Public Health, Karolinska Institutet, Stockholm, Sweden

* dalberghina@unime.it

**Data Availability Statement:** All relevant data are within the paper and its Supporting Information files.

## Abstract

Behavioral development in domestic dogs has been investigated for predicting suitability for service dog work or for matching with the "right" families as well as for identifying predispositions to behavioral problems. Findings from the scientific literature seem to confirm that conducting behavioral tests at 7 weeks of age is too early to reliably predict the temperament and personality of a dog. However, this period for domestic dogs is sensitive for early life learning and conditions during this time could have important consequences in adulthood.

The aims of this study were to evaluate inter-rater reliability of a simple standardized test and to investigate which factors influence the behavioral reaction of puppies. 105 seven-week old puppies were exposed to five subtests: social attraction, following, retrieving, sudden appearance, noise. During each task, the behaviour of each pup was scored on a 3–5 point scale that reflected the suitability of the pup's reaction to the task. Scores were evaluated for a single subtest and for two aggregate indicators (i.e. *response to a person*: social attraction subtest and following subtest and *response to object and noise*: retrieving subtest, sudden appearance subtest and noise subtest). Three assessors independently scored the dogs' reactions for each task. Inter-rater reliability of the three assessors were analyzed with Fleiss' Kappa and Kendall's coefficient, which showed a high inter-rater reliability in 4 of 5 tasks. The ordered logistic regression was carried out to obtain a proportional odds model that was used to model the relationship between sex, litter size, stimulating environment, parity of mother, adequate maternal behavior and high scores. Litter size and maternal parity were associated with test performance in *response to a person*. The variance of effect of litter was high in *response to object and noise*. Taken together, our results suggest that using this scoring system there is sufficient inter-rater reliability in the test and litter size and mother experience influences task performances related to dog-human interaction.

**Funding:** The authors received no specific funding for this work.

**Competing interests:** The authors have declared that no competing interests exist.

## Introduction

Behavioral development in domestic dogs has been investigated for matching puppies with the right families, identifying predispositions for behavioral problems at an early stage, and predicting suitability for working-dog organizations, which select dogs at a young age to train as service dogs (e.g. guide dog, hearing dog, medical alert dog, etc . . .). Puppy tests are typically aimed at investigating a variety of behavioural predispositions and often include interactions with unfamiliar people, play, exploration of novel environments or objects, and startle stimuli [1]. Puppy tests involve presenting a selection of tasks to puppies in a standardized manner, to allow for comparisons [2]. The period between 6 and 7 weeks of development may facilitate certain testing, since puppies haven't fully developed the fear imprinting response and they can be more easily handled by unknown people [3]. The potential for evaluation of dog-human relationships or the predisposition to learn from humans at this age could provide insights for improving adoptions since the recommended age for putting puppies up for adoption are around 8 weeks of age [4]. Although weaning may occur from 4 to 6 weeks of age, a puppy should never be adopted before 7.5 to 8 weeks of age since clinical observations indicate that the interaction occurring within the litter at this time and the effect of the mother are critical to a puppy's development, and early removal from the litter may result in emotional instability [5].

There are still concerns over the lack of standardisation amongst research on dog behavioral tests [6]. Published literature on puppy tests reveals that there has been little consistency in the tasks used, the age of testing and the form of evaluation or validation [7]. Furthermore, the predictive validity of early tests for predicting specific behavioral traits in adult pet dogs is limited [1].

The behavioral assays of the puppy tests developed by Campbell and by Jack and Wendy Volhard are the most commonly used in practice for seven-week-old puppies [8]. The Volhards' puppy aptitude test comprises of 10 subtests that incorporates tasks from the Campbell test [9] and from the Puppy Temperament Test [10]. It also includes an additional three tasks to test responses to touch, sound and the sudden opening of an umbrella [8]. In each subtest, puppies are scored on a scale from 1 to 6 depending on their behavioural response. This test has been rarely utilized in the scientific literature likely because the scoring method cannot be statistically analyzed but see Goleman et al. [11], Asher et al. [7], Majecka et al. [8] for a modified version of the test.

In the present study, we employed five subtests from the Volhard test: social attraction, following, retrieving, sudden appearance and noise. These subtests were selected for evaluating the following: "a response to a person" (by the social attraction and following subtests), "a response to object and noise" (by the retrieving, sudden appearance and noise subtests). For this simplified test version, we implemented a new scoring protocol that allowed for statistical analysis. We investigated the extent to which different observers describe the same individual the same way (inter-rater reliability) of the test and whether breed size, sex, litter size, maternal parity and care levels as well as environmental differences affects the behavior of seven-week-old puppies in these subtests.

## Material and methods

### Ethics statement

Special permission for use of animals (dogs) in this kind of behavioural study is not required in Italy. All procedures were performed in full accordance with Italian legal regulations (National Directive n. 26/14—Directive 2010/63/UE) and the guidelines for the treatment of

animals in behavioral research and teaching of the Association for the Study of Animal Behavior (ASAB). A written consent to video-record and use data in an anonymous form was obtained by the breeders prior to testing.

## Subjects

A total of 105 puppies (52% males and 48% females) from 21 litters belonging to 13 breeds were included in this study. Breeds were classified into 4 groups according to the expected mean adult body weight: small (10 kg and less), medium (between 11 to 25 kg), large (26 to 45 kg) and giant (over 45 kg) breeds (Table 1). The litter sizes varied from 1 to 13 (average 5.23 ± 2.88 Standard Deviation).

All puppies were tested at the breeders at the beginning of seven weeks of age (range 49–52 days). Information about maternal experience (36% primiparous or 64% multiparous) was collected directly by asking the breeders. Presence of adequate maternal behaviour was collected by asking the breeders if they observed "mother-pup interaction during feeding sessions, licking, contact, play, movement towards and away from the puppy" and their response, yes or not, was classified respectively as adequate (90%) or inadequate (10%) maternal behaviour, while information about environment was evaluated by direct inspection. Environment was classified as "stimulating" (42% of total observations) when kennels were located in the breeder's house, where puppies and their mother were exposed to all the stimuli of a typical household. In contrast, environment was classified as "not stimulating" (58% of total observations) when kennels were located outside of the breeder's house with limited human contact.

Puppies were tested individually away from conspecifics and were tested prior to their normal feeding time in the late afternoon between 4.00 and 6.00 pm. Testing took place between December 2017 and February 2019. Involved subjects were not housed for use in further research.

## Procedure

Tests were carried out in an environment unfamiliar to the puppies at the breeders' homes. All tests were conducted by the same examiner, who was unfamiliar to the puppies prior to the test. A second person filmed the test for subsequent video analysis. The reaction of the puppy was video recorded during five subtests. A description of each subtest along with the scoring

Table 1. Puppies used in the study: Size, breed, number of litters and gender (female and male).

| Size | Breed | Number of subjects | Litters | Female | Male |
|---|---|---|---|---|---|
| Small | French Bulldogue | 9 | 2 | 3 | 6 |
| Small | Cavalier King Charles Spaniel | 5 | 2 | 2 | 3 |
| Small | Pomeranian | 4 | 2 | 1 | 3 |
| Medium | Chow Chow | 4 | 1 | 2 | 2 |
| Medium | Bull Terrier | 7 | 2 | 2 | 5 |
| Medium | Belgian Sheepdog (Groenendael) | 3 | 1 | 2 | 1 |
| Large | American Akita | 13 | 2 | 7 | 6 |
| Large | Cane Corsos | 5 | 1 | 2 | 3 |
| Large | Mannara's Dog | 10 | 2 | 5 | 5 |
| Large | Labrador Retriever | 9 | 2 | 5 | 4 |
| Giant | Caucasian Dog | 7 | 1 | 3 | 4 |
| Giant | Saint Bernard | 23 | 2 | 12 | 11 |
| Giant | Bernese Mountain Dog | 6 | 1 | 4 | 2 |

**Table 2. Behavioral response and scoring protocol for the 5 subtests.** All puppies were individually recorded under each test in a novel space. Video recordings of their behavioral responses in each subtest were independently scored by three assessors using a scale of 60/75/100 to 300.

| Subtest | Response | Score |
|---|---|---|
| Social attraction | Puppy doesn't come at all or moves away in another direction | 60 |
| | Puppy starts to come but changes direction or stops | 120 |
| | Puppy comes after having gone to another direction | 180 |
| | Puppy goes to the examiner hesitantly | 240 |
| | Puppy goes to the examiner immediately | 300 |
| Following | Puppy stays on the same place / moves away in another direction | 100 |
| | Puppy follows the examiner hesitantly/follow immediately the examiner, tail straight up and nibbles them | 200 |
| | Puppy follows the examiner readily | 300 |
| Retrieving | Puppy doesn't chase object | 60 |
| | Puppy starts to chase object, but it loose interest | 120 |
| | Puppy chases object, picks it up and runs away | 180 |
| | Puppy chases object and returns without it to the tester | 240 |
| | Puppy chases object, picks it up and returns with it to tester | 300 |
| Sudden Appearance | Puppy runs away/Puppy looks and runs to umbrella, mouthing or biting it | 75 |
| | Puppy looks at umbrella in an excited way (wagging his tail) but doesn't approach to it | 150 |
| | Puppy moves to umbrella and attempts to investigate by teeth | 225 |
| | Puppy moves toward the umbrella and investigates in a excited way (sniffing and wagging his tail) | 300 |
| Noise | Puppy ignores the sound/Cringes, backs off | 100 |
| | Puppy listens and detects sound but doesn't move to the source | 200 |
| | Puppy listens, detects sound and moves to the sound source | 300 |

protocol is in Table 2. Each puppy received the 5 subtests in the same order and each test lasted about five minutes per puppy.

During the testing phase, puppies were evaluated on how they responded to a person under the social attraction subtest and the following subtest (Fig 1). Puppies were also evaluated on their response to a novel object and sudden noise in the retrieving subtest, sudden appeerence subtest and noise subtest (Fig 2). The behaviour of puppies for each subtest was scored on a scale of 60 to 300 where a higher score represented a better response and/or behavior. As the

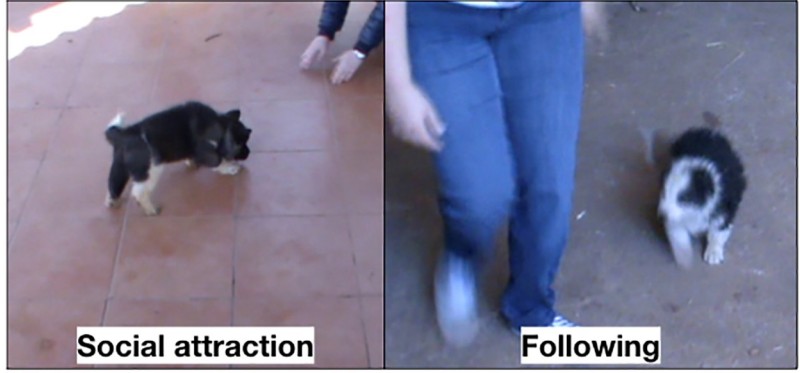

**Fig 1. Social attraction and following subtests performed respectively by an American Akita and a Mannara's dog puppy.**

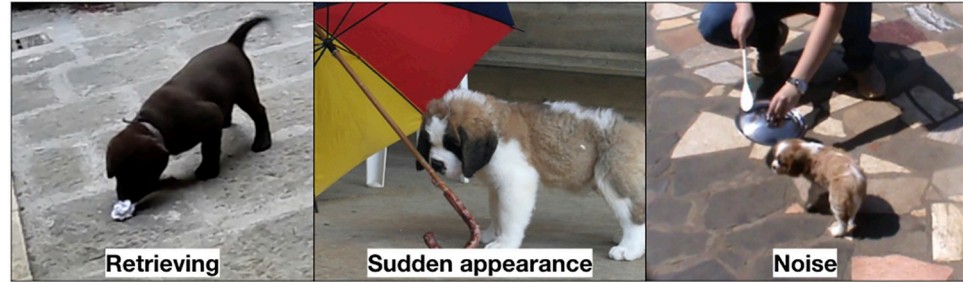

**Fig 2. Retrieving, sudden appearance and noise subtests performed respectively by a Labrador, a Saint Bernard and a Cavalier King Charles Spaniel puppy.**

maximum score was 300, the score assigned to the behavioral response for each test was obtained by dividing 300 by 5, 4 or 3 depending of the number of observed reactions. Three assessors independently examined each video and scored each puppy's behavioral response in each subtest using the scoring protocol described in Table 2.

**Social attraction subtest.**   Puppy is placed in the test area. Examiner kneels down and coaxes the puppy to come to them with encouragement and gently clapping hands.

**Following subtest.**   Examiner stands up and slowly walks away encouraging the puppy to follow.

**Retrieving subtest.**   The examiner crouched next to the puppy and attracts its attention with a crumpled piece of paper. When the puppy shows interest, the tester rolls the paper a small distance from the puppy, encouraging it to pick up the paper.

**Sudden appearance subtest.**   The examiner calls the puppy and, when it reaches a distance of 1 m, opens an umbrella and drops it immediately on the ground.

**Noise subtest.**   The puppy is placed in the center of the testing area and the examiner, stationed at the perimeter, makes a sharp noise by banging a spoon on a pan.

## Statistical analysis

The results were analyzed to identify inter-rater reliability and the main factors associated to higher scores. Influence of observer was evaluated by variability of assigned scores in response to subtests. Fleiss' Kappa (K) and Kendall's coefficient (Kendall's W) were calculated for assessing the reliability of agreement between raters [12, 13]. Mean scores were analyzed by the proportional odds model for ordinal logistic regression where single variables for each subtest (sex, litter size, maternal parity and good maternal care, environment) were specified in an additive model. The proportional odds model is a class of generalized linear models used for modeling the dependance of an ordinal response on discrete or continuous covariates. Kendall's Tau-b was used to correlate size with litter size. Since a high correlation, b = 0.63 P<0.001, was found, we chose to use litter size as a covariate. Scores were evaluated for each single subtest and for two main types of responses (*response to a person* and *response to object and noise*). For these types of responses, mean scores were calculated from the mean of each subtest, i.e. mean scores obtained by 3 observers for the social attraction test + mean scores for the following test were used for "response to a person". Scores were classified as low (mean 100), medium (mean 200) and high (mean 300). Due to the small sample size P values <0.10 were considered significant. R (3.3.2) statistical programs were used for all the analysis.

**Table 3. Standard deviation of scores between observers and K Fleiss and Kendall's W values.**

| Test (categories distance) | SD | K | W |
|---|---|---|---|
| Social attraction (60) | 24.92 | 0.55 | 0.86 |
| Following (100) | 26.5 | 0.57 | 0.83 |
| Retrieving (60) | 7.68 | 0.70 | 0.89 |
| Sudden appearence (75) | 20.71 | 0.44 | 0.76 |
| Noise (100) | 18.18 | 0.67 | 0.84 |

## Results

### Reliability between observers

Standard deviation between observers was lower than distance between categories. Table 3 shows K and Kendall's W (from +1: complete accordance to -1: complete disaccordance) for each stimulus. Lower values were found for the sudden appearance subtest while higher values were found for the retrieving test and noise subtest.

### Proportional-odds cumulative logit model

Table 4 shows results from the proportional-odds cumulative logit model for each subtest. Variance due to the litter was very low in all tasks except the sudden appearance and noise subtests. Litter size significantly influenced the social attraction (P = 0.03), following (P = 0.06) and sudden appearance (P = 0.06) subtests. Scores of each single subtest were added for each aggregate indicator ("response to a person" and "a response to object and noise") and mean scores were used as follows for evaluation: 300 (high), 200 (medium) and 100 (low). Table 5 shows results from the proportional-odds cumulative logit model for each indicator. "Response to a person" was significantly influenced by litter size (P = 0.02) and mother parity (P = 0.06). In "Response to object and noise" indicator the variance of litter was high. As shown in Fig 3, for the response to a person indicator, puppies from small litters, as well as puppies from multiparous mother, showed a tendency to have higher scores than others.

## Discussion

We investigated inter-rater reliability and influencing factors of a simple puppy test. Though only three observers scored puppy behaviour, the test categories clearly represented puppy behaviour (Kendall's W very high). Rating of individual tests showed a high degree of concordance except for the sudden appearance subtest. Although this subtest needs some adjustments to its protocol, the low variability and high values of Kendall's W validates the credibility of this test. The inter-observer variability among the three observers was low. There is little reporting of inter-rater reliability agreement across two or more independent observers [6]. These findings indicate that using this scoring system for the same puppy could be evaluated similarly by different individuals. Inter-rater reliability is a critical assurance that the scoring system is well defined and can be replicated [14]. In order to better standardize the test, adjustments to the scoring protocol of the sudden appearance subtest are recommended. It is recommended that there should be no distinction between the behavioral responses: "Puppy moves to umbrella and attempts to investigate by teeth" and "Puppy moves toward the umbrella and investigates in a excited way (sniffing and wagging his tail)". This is because for puppies at 7 weeks of age, using their mouths is their main way to interact with everything in their environment. Puppies start to learn bite inhibition while with their littermates. Thus, we would

**Table 4. Estimate of effects on proportional odds cumulative logit scale, SE, P and 90% Confidence Intervals for each subtest.**

| | ($\alpha = 0.10$); β log odds. | | | |
|---|---|---|---|---|
| **Social attraction** Variance in the logit of the scores due to the litter = 1.2e-08 | **β** | **SE (β)** | **P-value** | **Confidence Interval (90%)** |
| Intercept (log odds of lower scores) | -0.37 | 1.64 | 0.82 | [-3.06, 2.33] |
| Litter size | 0.17 | 0.08 | 0.03 | [0.04, 0.29] |
| Suitable Environment | 0.03 | 0.43 | 0.95 | [-0.68, 0.74] |
| Sex (Male) | 0.06 | 0.37 | 0.88 | [-0.55, 0.66] |
| Mother (adequate maternal behaviour) | -0.83 | 0.81 | 0.30 | [-2.16, 0.50] |
| Multiparous mother | 0.48 | 0.38 | 0.20 | [-1.09, 0.14] |
| **Following** Variance in the logit of the scores due to the litter = 2e-08 | **β** | **SE (β)** | **P-value** | **Confidence Interval (90%)** |
| Intercept (log odds of lower scores) | 0.35 | 1.69 | 0.73 | [-2.43, 3.13] |
| Litter size | 0.14 | 0.08 | 0.06 | [0.01, 0.26] |
| Suitable Environment | 0.10 | 0.44 | 0.83 | [-0.63, 0.82] |
| Sex (Male) | 0.23 | 0.38 | 0.55 | [-0.40, 0.86] |
| Mother (adequate maternal behaviour) | -0.77 | 0.85 | 0.36 | [-2.16, 0.62] |
| Multiparous mother | -0.46 | 0.40 | 0.26 | [-1.12, 0.21] |
| **Retrieving** Variance in the logit of the scores due to the litter = 1.2e-07 | **β** | **SE (β)** | **P-value** | **Confidence Interval (90%)** |
| Intercept (log odds of lower scores) | -1.20 | 1.78 | 0.48 | [-4.13, 1.74] |
| Litter size | 0.07 | 0.07 | 0.34 | [-0.05, 0.19] |
| Suitable Environment | -0.37 | 0.44 | 0.40 | [-1.09, 0.35] |
| Sex (Male) | -0.21 | 0.38 | 0.58 | [-0.84, 0.42] |
| Mother (adequate maternal behaviour) | 0.35 | 0.87 | 0.69 | [-1.09, 1.78] |
| Multiparous mother | 0.21 | 0.39 | 0.59 | [-0.43, 0.86] |
| **Sudden appearence** Variance in the logit of the scores due to the litter = 0.59 | **β** | **SE (β)** | **P-value** | **Confidence Interval (90%)** |
| Intercept (log odds of lower scores) | 3.30 | 2.00 | 0.10 | [-0.02, 6.56] |
| Litter size | -0.22 | 0.12 | 0.06 | [-0.41, -0.03] |
| Suitable Environment | -0.75 | 0.61 | 0.22 | [-1.76, 0.26] |
| Sex (Male) | -0.17 | 0.40 | 0.67 | [-0.83, 0.48] |
| Mother (adequate maternal behaviour) | -1.22 | 0.96 | 0.20 | [-2.80, 0.36] |
| Multiparous mother | -0.04 | 0.56 | 0.94 | [-0.97, 0.89] |
| **Noise** Variance in the logit of the scores due to the litter = 0.48 | **β** | **SE (β)** | **P-value** | **Confidence Interval (90%)** |
| Intercept (log odds of lower scores) | 1.4 | 2.0 | 0.48 | [-1.86, 4.59] |
| Litter size | 0.10 | 0.11 | 0.3 | [-0.08, 0.29] |
| Suitable Environment | -0.85 | 0.63 | 0.2 | [-1.88, 0.18] |
| Sex (Male) | -0.26 | 0.41 | 0.5 | [-0.93, -0.41] |
| Mother (adequate maternal behaviour) | -0.94 | 0.97 | 0.3 | [-2.54, 0.65] |
| Multiparous mother | 0.07 | 0.57 | 0.9 | [-0.86, 1.00] |

propose to remove the third behavioral response and keep the remaining three behavioral responses for the sudden appearance subtest.

In the model, six covariates are included as predictive factors of high subtest scores: sex, litter size, stimulating environment, parity of bitch and maternal care levels. The results of the regression in Table 5 show that litter size and mother parity are found to be significant predictors of behavioral response. Litter size is a complex factor that is related to genetic and environmental factors. For instance, a puppy that interacts with more conspecifics during the socialization period can develop differently than a puppy with few or no interactions with conspecifics. Previous research on a population of German Shepherds reported that factors such as litter size, sex ratio, growth rate and season of birth can significantly affect behaviour [15].

**Table 5. Estimate of effects on proportional odds cumulative logit scale, SE, P and 90% Confidence Intervals.**

| | (α = 0.10); β log odds. | | | |
|---|---|---|---|---|
| **Response to a person** Variance in the logit of the scores due to the litter = 6.2e-09 | **β** | **SE (β)** | **P-value** | **Confidence Interval (90%)** |
| Intercept (log odds of lower scores) | -0.26 | 1.28 | 0.84 | [-2.37, 1.84] |
| Litter size | 0.15 | 0.06 | 0.02 | [0.04, 0.25] |
| Mother (adequate maternal behaviour) | -0.21 | 0.66 | 0.75 | [-1.29, 0.87] |
| Multiparous mother | -0.73 | 0.39 | 0.06 | [-1.37, 0.10] |
| Response to object and noise Variance in the logit of the scores due to the litter = **0.23** | **β** | **SE (β)** | **P-value** | **Confidence Interval (90%)** |
| Intercept (log odds of lower scores) | 1.41 | 1.81 | 0.44 | [-1.57, 4.38] |
| Litter size | 0.03 | 0.09 | 0.73 | [-0.12, 0.19] |
| Suitable Environment | -0.88 | 0.54 | 0.10 | [-1.75, 0.005] |
| Sex (Male) | 0.07 | 0.40 | 0.87 | [-0.59, 0.73] |
| Mother (adequate maternal behaviour) | -0.74 | 0.87 | 0.39 | [-2.17, 0.68] |
| Multiparous mother | 0.19 | 0.49 | 0.70 | [-0.62, 0.99] |

Litter size also influences the weight and growth of puppies during the first month of age [16]. Weight has an impact on health and on certain behavioral aspects [17], with larger female puppies being more active and explorative than their smaller counterparts when subjected to a behaviour test at 8 weeks of age [16]. Litter size also influences behavioral responses in other altricial species [18, 19]. In a small litter, the possibility of physical contact between the mother and any sibling is greater, which could be related to better performance on behavioral tests.

Levels of canine maternal care have been reported to affect performance on a temperament test that is conducted at 15–18 months of age [20]. A questionnaire-based study asked dog

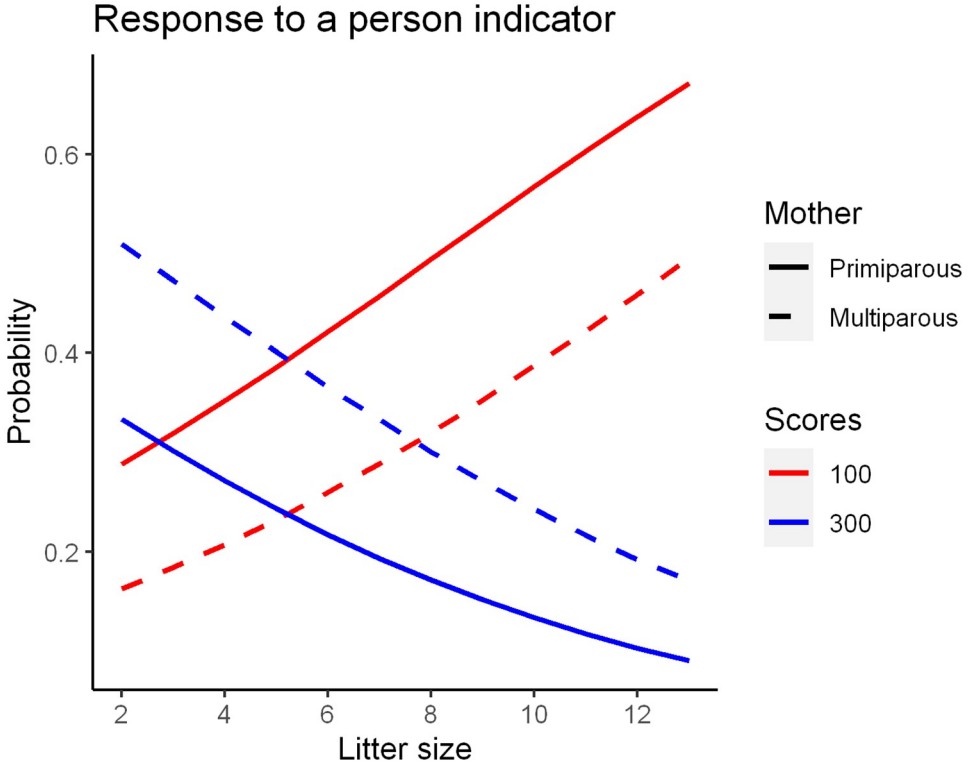

**Fig 3. Influence of litter size and mother parity on "response to human" indicator.**

owners to grade the quality of maternal care, specified as spending time with and taking care of the pups. Lower scores, indicating an estimated poor quality of maternal care by the owner, were associated with fearful behaviour in the adult dog [21]. A longer daily duration of maternal care during the first three weeks postpartum was associated with more exploratory behaviour and less signs of stress in eight week old puppies [22] however, there is no data available for younger ages. Furthermore, a recent review of current literature confirms that the behaviour of an adult dog is determined to a large extent by the quality of maternal care, its attachment style to its mother, and the variety of both social and non-social stimuli provided during the early and late socialisation period [23]. Unlike the reports in the literature, our results did not show an effect of adequate maternal behaviour on performance response to subtests. This could be due to the subjectiveness of the breeders asked to judge this behaviour. For future studies, maternal behaviour should also be recorded directly to determine whether it has an impact on responses to these test. More specifically, in order to have a better standardization of observed behaviours, maternal behavior should be recorded 1 day per week continuously every second hour over a 24-hour period during the first 3weeks postpartum as described by Foyer et al. [20]

In this study, puppies raised by experienced mothers had a tendency to perform better in the subtests related to "response to a person". In a previous study, mother parity influenced the behavior of puppies of different ages in some subtests [15, 16]. Furthermore, any change in maternal behaviour due to previous reproductive experience may affect the behaviour of puppies. Evaluation of the records of German shepherd dogs from the Swedish armed forces demonstrated that puppies from more experienced bitches scored better for confidence and physical engagement, when tested as young adult dogs [15]. The effect of parity should therefore be further explored in future studies.

Our results showed a trend for suitable environment having an impact on response to the object and noise aggregate indicator (P = 0.1). Dogs raised in domestic environments were less likely to develop fear and aggression towards unfamiliar people compared to dogs raised in non-domestic environments [24]. Sufficient exposure to relevant stimuli during the early socialisation period appears to be associated with lower fearfulness and aggression in dogs [25].

Our simplified version of the Volhards test is quick, easy to administer, free from physical discomfort situations and feasible in different environments. We recommend testing puppies at seven weeks of age because this age is right in the middle of the socialization period and right before the fear period (which can start at about 8 weeks of age, but also varies age between breeds and individuals) [5]. In future studies of puppy behavioral tests, we recommend including litter size and parity of mother as predictors of scores in the "response to a person" subtests. We identified variability in our data due to a litter effect in the "response to object and noise" subtests. Litter effect should always be taken into consideration because differences in behavior among individuals may arise from a common litter environment or from hereditary factors [8]. Due to the small sample size and large heterogeneity of the factors these results have to be considered as preliminary findings. In our study, all breeds, even with size differences, were treated as a single breed because our data were not adequately balanced to include breed as a factor.

## Conclusion

Our results support the hypothesis that specific context influences the performance of seven-week-old puppies on a behavioural test. This study shows that there is sufficient inter-rater reliability in the test. The designed scoring system for the test can be used reliably different

people and for quantitative analysis. Further work is needed to determine if performances difference factors remain consistent when dogs are retested at a later stage.

## Supporting information

**S1 Data.**
(XLSX)

**S1 Fig.**
(PNG)

## Acknowledgments

We wish to thank the breeders for their cooperation in conducting this research. We would like to thank Dr. Winnie Y. Chan for her advice and help in English correction and Dr. Alessandra Statelli for her assistance with manuscript preparation. We also thank the anonymous reviewers for their comments and suggestions to improve the quality of this article.

## Author Contributions

**Conceptualization:** Daniela Alberghina, Michele Panzera.

**Data curation:** Daniela Alberghina, Mauro Gioè.

**Formal analysis:** Mauro Gioè.

**Investigation:** Fabiola Giunta.

**Supervision:** Michele Panzera.

**Writing – original draft:** Daniela Alberghina.

**Writing – review & editing:** Daniela Alberghina.

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
