## [Decision Letter · Decision Letter 0]

27 Mar 2020

PONE-D-20-02339

Development of a simple standardized test to evaluate influencing factors on the behavior of seven week puppies (Canis familiaris): a preliminary study

PLOS ONE

Dear Dr Alberghina

Thank you for submitting your manuscript to PLOS ONE. After careful consideration, we feel that it has merit but does not fully meet PLOS ONE’s publication criteria as it currently stands. Therefore, we invite you to submit a revised version of the manuscript that addresses the points raised during the review process.

Many thanks for submitting your manuscript to PLOS One

Your manuscript was reviewed by two experts in the field. they have both provided a large number of comments and concerns about the manuscript which need to be addressed prior to acceptance

As both reviewers see value in the work, I have given you the opportunity to work on the comments suggested and to resubmit it for re-review

Please ensure that you write a detailed response to reviewers, covering each of their points raised

I wish you the best of luck with your revisions

Hope you are keeping safe in these difficult times

Thanks

Simon

We would appreciate receiving your revised manuscript by May 11 2020 11:59PM. To enhance the reproducibility of your results, we recommend that if applicable you deposit your laboratory protocols in protocols.io, where a protocol can be assigned its own identifier (DOI) such that it can be cited independently in the future. For instructions see: http://journals.plos.org/plosone/s/submission-guidelines#loc-laboratory-protocols

We look forward to receiving your revised manuscript.

Kind regards,

Simon Russell Clegg, PhD

Academic Editor

PLOS ONE

Journal Requirements:

2. In your Methods section, please provide additional details regarding the source of the dogs used in your study and ensure you have described where the dogs were obtained from.

3. We noticed you have some minor occurrence of overlapping text with the following previous publications, which needs to be addressed:

https://journals.plos.org/plosone/article?id=10.1371/journal.pone.0101237

https://brill.com/view/journals/beh/155/2-3/article-p83_1.xml?language=en

In your revision ensure you cite all your sources, and quote or rephrase any duplicated text outside the methods section. Further consideration is dependent on these concerns being addressed.

Reviewers' comments:

Reviewer's Responses to Questions

**Comments to the Author**

1. Is the manuscript technically sound, and do the data support the conclusions?

Reviewer #1: Partly

Reviewer #2: Partly

2. Has the statistical analysis been performed appropriately and rigorously? 

Reviewer #1: No

Reviewer #2: No

3. Have the authors made all data underlying the findings in their manuscript fully available?

Reviewer #1: No

Reviewer #2: No

4. Is the manuscript presented in an intelligible fashion and written in standard English?

Reviewer #1: No

Reviewer #2: No

5. Review Comments to the Author

Reviewer #1: I am afraid this manuscript needs quite a bit of extra work before I would consider it to be publishable. The methods are not described in enough detail, I am not convinced that the statistical analysis is appropriate as (no control for dogs being of the same litter), the results are not fully reported (only ‘significant’ statistics are given) and the findings are overstated. I’ve provided specific comments below:

The abstract describes part of the aim as developing a test, but this isn’t what the authors did; they used an existing test and looked at factors that were associated with different scores on the test plus inter-observer reliability. Please make it clearer what the study actually did.

Line 33: the effect sizes for breed-group were small, you don’t have grounds to claim ‘considerable influence’ on the test performance. There was some association at best.

Line 47: “may facilitate certain testing since puppies are motivated to approach unknown people” I’m not sure what point you are making here. Do you mean that because the fear response hasn’t fully developed yet that they can be more easily handled by strangers to conduct the test?

Line 49: remove ‘stage of’ so it reads “at this age”

Line 54-56: I have problems with this whole section that it overstates the impact of genetics on behaviour. What we know from research is that dogs are individuals, and breed has little to do with general personality – hence the large amount of within-breed variance. Only around 30% of personality is heritable, of which a large part is probably due to shared early environment (including the womb environment). This bit in particular needs to be written with more caution. Only some of what determines a dog’s personality is affected by its genetics, and very little by its breed. There is a lot of misunderstanding rife in the public about how much breed influences dog personality and as dog scientists we must be very careful we do not fuel this further. When testing puppies at such a young age, estimates of heritability are stronger as variability is reduced, but this is to be expected since the dogs have all shared the same life experience so far. When heritability of behaviour (and variance) is examined in older dogs, once environment has had a chance to influence them, it drops dramatically showing that early estimates are over-estimates. This is an inherent problem with drawing conclusions about genetic influence from variability shown under limited, standardised conditions.

Line 89: what information did you collect about maternal care levels and how?

Line 91 & 93: ‘stimulating’ and ‘non-stimulating’ both are 42%, so what were the remaining 16%?

Line 112: this makes no sense “Scoring 300 the most suitable reaction, interval between behavior options for each subtest were obtained dividing by 5, 4 or 3 depending of the number of observed reactions”. Please re-phrase this.

Statistical analysis: please use inter-observer reliability, instead of ‘variability’ as the two terms mean different things, and reliability is the most commonly used/understood. This also relates to your Abstract, where you state inter-observer variability as being low, but it would be much easier to understand if you said inter-observer reliability was high. Please also state here what cut-offs you were prepared to consider acceptable in the K and W statistics: these decisions should always be made before you run your analysis. How were the scores distributed? Did you check for normality first? Since you’re working with dogs from litters, the individual dogs are not independent of each other so ‘litter’ should be taken into consideration as a random effect in any analysis. As far as I can tell you did not control for litter in this analysis, which means you have not adjusted for pseudo-replication that using dogs of the same litter would cause. How were the ‘complex indicators’ calculated? Please describe this. Considering the large number of comparisons made here I would also expect to see P-value correction to account for multiple testing.

Table 3 & 4 should be merged into one. You should also provide the confidence interval for the K and W coefficients.

Table 5 must report all values – not just ‘significant’ ones. It’s also important to include the confidence interval around the Beta estimate, as p-values alone are essentially meaningless.

Line 165: The W values were high, but not ‘very high’ (>0.90 would be very high).

Line 170: typo – ‘avaluated’

Line 175: typo – ‘indipendently’

Line 180: repeated word “in test social attraction test”

Line 203: What does this mean: “positive influence on “learning predisposition” indicator in pups”?

You say in the abstract that “In order to better standardize the test, adjustments to the scoring protocol are recommended yet I can’t find any such recommendations anywhere.

There’s one sentence on the study limitations, which isn’t really good enough.

Reviewer #2: The current manuscript examines responses of 7-week old puppies to a series of behavioural tests, examining both interobserver reliability for scoring for the tests, and the effects of various puppy-related and environment-related variables on their responses. Examining early factors and their influence on puppy and later adult behaviour is an important topic. However, I have some questions about the overall objective of the current work as well as the approach that was taken. I have summarized these concerns below, and provided specific relevant comments below in the ‘general comments’ section

1. Based on the information provided in the introduction, the rationale for this particular study is unclear and needs strengthening. Why do we need to understand how different factors influence the behaviour of young puppies when previous studies have shown that performance on tasks at this age has poor predictive power for later behaviour? I recommend that the authors reframe the introduction to clearly outline what research on puppy testing has already been published and why this particular study is necessary and important – I’m assuming it is to understand sources of variability so that test performance can be improved, but this isn’t clear from what is written.

2. It’s unclear why the authors selected to use these particular tests for testing the puppies, so the justification needs to be strengthened. Why are these areas of assessment important, and why did they select this particular test when it hasn’t been used in any previously published work? Also, the methods that were used for testing are not described in sufficient detail for proper assessment of rigour and for replication, and some of the descriptions for scoring in Table 2 are quite vague. Were these the final descriptions that were used for the observers during scoring?

3. The methods and results for the regression analysis are a bit confusing and this makes it difficult to properly assess the findings and conclusions for this study. Areas where more information is needed are described in more detail below. However, based on the information provided it appears that the analysis doesn’t account for clustering, which is critical when examining litters of puppies.

General Comments

Data statement: The authors have indicated that all data are present in the manuscript, but only aggregate data is presented. The raw data is not available for review.

The manuscript requires further editing for grammar and spelling, particularly the discussion.

L16-19: The aims for the current study are unclear. If 7 weeks of age is too young to predict later temperament, why is it important to understand factors that influence behaviour at this age? Clarification is needed

L20: replace stimuli with tests or tasks

L21-22: I wonder if the suitability of the response depends on the purpose for which the puppy has been bred? Is there a way to word this that is descriptive rather than suggesting that a particular response is best?

Complex indicators: What is the rationale for combining particular tasks together into complex indicators. Is there some indication that responses on these tasks are related?

L27-30: The results for these complex indicators are not presented in the main body of the manuscript.

Introduction: The intro is a bit confusing because the authors start out discussing the potential for early testing to predict later potential, but this does not actually relate to their objective of looking at factors that influence the behaviour of puppies during early behaviour testing. The rationale for this particular study needs to be more explained more clearly.

L44-46: The following sentences aren't really necessary for the introduction: "Analyses of behaviour normally involves measuring frequency, duration and latency of specific behaviours [3]. Rater-coding is normally done on a predetermined scale that may have only 3, 5, or 7 points [4]."

L47: Can you clarify why increased motivation to approach might be useful for these assessments?

L50-53: Given the objective of this study, it is important to summarize the literature on puppy assessments that have been published to date. What have people tried, what did they find and why is it inadequate and requiring further study? There is quite a large literature on this topic (both general and applied to selection for certain programs) and few studies are cited above but very little detail is provided to develop the rationale for the current study.

L60-61: Please clarify that these tests are used in practice for assessment, and that these aren't methods that have been used and assessed in the literature. What is your rationale for using these tests rather than other tests that have been used and assessed previously in the peer-reviewed literature?

L63: Missing space in "outthe"

L63-65: Is it necessary to provide this information when the authors chose not to use these categories? Instead please provide further justification for the approach that you chose to go with (ie, genetic relatedness). If you do decide to keep this information in the paper, please state clearly why this approach is inappropriate for your purposes.

L69-70: The objectives for the current study could use some clarification. What was your rationale for testing these specific factors, and did you have any predictions about how these factors would affect puppy behaviour beforehand?

L74: Add an 's' to ethics

L85: Add an 's' to breeds

L89-90: How, specifically, was the level of maternal care evaluated in the current study?

L91-93: This only accounts for 84% of observations - how were the others classified?

L95-96: Were the puppies tested at a particular time of day?

Table 1: Please expand this table to list the specific litters and number of puppies per litter to provide a better representation of the sample.

L110-113: I'm having trouble understanding the methods described here - please clarify by providing further details

Table 2: The rows on the table don't line up properly so it is difficult to determine which subtests the responses align with.

Table 2: Fix spelling for Appearance

Table 2: Since this isn't a series of tests that has been previously published in the literature, please describe the methods in sufficient detail that they can be replicated.

Table 2 - Sudden Appearance Test: What does curiosity refer to specifically? And for the final category, what constituted investigation?

Table 2 - Noise test: Again, what specific behavioural responses was indicative of curiosity?

L126-127: Please provide basic information and a reference for interpretation of these values. There are standard cutoffs for interpretation that are commonly used in the literature.

L127-129: Further details are needed on the methods used for statistical modelling. Does mean score refer to the mean of all three observers, and/or were the scores converted in some manner to categories prior to ordinal logistic regression? Also, what methods, if any, were used to account for clustering with litter (e.g., was litter included as a random effect)? We can expect that litter effects are likely quite large, so realistically your sample size is reduced to the number of litters unless you can account for clustering in some way.

Table 3: Why were standard deviations calculated? This seems unnecessary when Kappa and Kendall's are presented.

L150-152: Please move this information to the M&M. Does this mean that scores were re-categorized as 0-100, 101-200, 201-300 prior to analysis? What was the rationale for this decision?

L153-157: The figures alone do not provide sufficient information for evaluating the results from these models. Please provide the relevant values to support the comparisons that are made within the text or in Table 5.

Table 5: What is your justification for using a significance level of 0.1? Info should be provided in M&M

Table 5: I'm struggling a bit to interpret the information provided for differences amongst breed groups. For example, what was the overall value for genetic group and which referent are these in comparison to? Or was each group looked at separately in comparison to others not in that group? Apologies if I'm misinterpreting how this was done but it isn't clear to me.

Table 5: Sudden appearance test – typo

L164: The agreement between observers suggests reasonable agreement for scoring, but I'm not sure how this relates to clear representations of puppy behaviour. Please explain

L165: What are your interpretations of these statistics based on - please provide references.

L170: While the Kendall's W values are good, the Kappa values are in the moderate agreement range. It might be worth discussing why both were used, and differences in interpretation between the two.

L189: Can you be more specific about what aspects of behaviour were affected in this previous study?

Discussion: The discussion might benefit from additional consideration of how the current puppy tests are similar/dissimilar from previous research in this area.

L215-216: The final sentence of the conclusion is not supported by the data presented.

L268: This reference is not available at the link provided.

Figure 6: Should the blue label read Unsuitable or Unstimulating?

6. PLOS authors have the option to publish the peer review history of their article (what does this mean?). If published, this will include your full peer review and any attached files.

Reviewer #1: No

Reviewer #2: No

---

## [Author Response · Author response to Decision Letter 0]

13 May 2020

We agree with all reviewer comments and we have revised our manuscript accordingly. We are including all reviewers’ suggestions and clarifying the text when needed. We are confident that the new version of the manuscript will be greatly improved. 

Reviewers' comments

Reviewer #1: I am afraid this manuscript needs quite a bit of extra work before I would consider it to be publishable. The methods are not described in enough detail, I am not convinced that the statistical analysis is appropriate as (no control for dogs being of the same litter), the results are not fully reported (only ‘significant’ statistics are given) and the findings are overstated. I’ve provided specific comments below:

The abstract describes part of the aim as developing a test, but this isn’t what the authors did; they used an existing test and looked at factors that were associated with different scores on the test plus inter-observer reliability. Please make it clearer what the study actually did.

We have modified the title of the manuscript as follow “Behavior test for seven-week old puppies (Canis familiaris): inter-rater reliability and factors associated with test performance ” and we have modified the abstract. We reanalysed our data and present the new results in the abstract and in text. 

Line 33: the effect sizes for breed-group were small, you don’t have grounds to claim ‘considerable influence’ on the test performance. There was some association at best.

We have decided to exclude the breed effect because as suggested by the reviewer, breed itself has little to do with general personality. Moreover, our sample was too small to evaluate differences between breeds. Instead, we reclassified dog breeds according to size as there is evidence for size-related differences in personality and reanalyzed our data. Using Kendall’s Tau-b a high correlation, 0.63, was found between breed size and litter size. 

Line 47: “may facilitate certain testing since puppies are motivated to approach unknown people” I’m not sure what point you are making here. Do you mean that because the fear response hasn’t fully developed yet that they can be more easily handled by strangers to conduct the test?

We have modified this sentence

Line 49: remove ‘stage of’ so it reads “at this age”

Amended

Line 54-56: I have problems with this whole section that it overstates the impact of genetics on behaviour. What we know from research is that dogs are individuals, and breed has little to do with general personality – hence the large amount of within-breed variance. Only around 30% of personality is heritable, of which a large part is probably due to shared early environment (including the womb environment). This bit in particular needs to be written with more caution. Only some of what determines a dog’s personality is affected by its genetics, and very little by its breed. There is a lot of misunderstanding rife in the public about how much breed influences dog personality and as dog scientists we must be very careful we do not fuel this further. When testing puppies at such a young age, estimates of heritability are stronger as variability is reduced, but this is to be expected since the dogs have all shared the same life experience so far. When heritability of behaviour (and variance) is examined in older dogs, once environment has had a chance to influence them, it drops dramatically showing that early estimates are over-estimates. This is an inherent problem with drawing conclusions about genetic influence from variability shown under limited, standardised conditions.

No genetic influence was considered in this new version of the manuscript

Line 89: what information did you collect about maternal care levels and how?

Maternal care levels were collected by asking the owners about licking, nursing, contact, play, movement towards and away from the puppy...

Line 91 & 93: ‘stimulating’ and ‘non-stimulating’ both are 42%, so what were the remaining 16%?

We have corrected the wrong percentage reported in the text!

Line 112: this makes no sense “Scoring 300 the most suitable reaction, interval between behaviour options for each subtest were obtained dividing by 5, 4 or 3 depending of the number of observed reactions”. Please re-phrase this.

We re-phrased this as follows: 

The most suitable reaction was scored 300, and the interval between behaviour options for each subtest were obtained by dividing by 5, 4 or 3 depending of the number of subtest options. 

Statistical analysis: please use inter-observer reliability, instead of ‘variability’ as the two terms mean different things, and reliability is the most commonly used/understood. This also relates to your Abstract, where you state inter-observer variability as being low, but it would be much easier to understand if you said inter-observer reliability was high.

We replaced the term variability in the abstract and in the text

 Please also state here what cut-offs you were prepared to consider acceptable in the K and W statistics: these decisions should always be made before you run your analysis. How were the scores distributed? Did you check for normality first? 

 We check for normality but data were not normal. K and W are non-parametric tests, then normality of scores is not required (neither for cumulative logit normality is required). K and W values were defined prior to running the statistical analysis and are defined based on the values reported in the literature (references have been included in the text). 

Since you’re working with dogs from litters, the individual dogs are not independent of each other so ‘litter’ should be taken into consideration as a random effect in any analysis. As far as I can tell you did not control for litter in this analysis, which means you have not adjusted for pseudo-replication that using dogs of the same litter would cause.

We reanalyzed our data with “litter” as a random effect and results were adjusted for pseudo-replication, we are grateful for suggestion. 

 How were the ‘complex indicators’ calculated? Please describe this. 

Description was added in the text:

Complex indicators were calculated from mean of each subtest as follows: mean scores obtained by observers for social attraction + following test for complex indicator “dog-human interaction”. Mean scores obtained by observers for the sum of other subtests for complex indicator “learning predisposition”. Scores were considered low (mean 100), medium (mean 200) and high (mean 300). 

Considering the large number of comparisons made here I would also expect to see P-value correction to account for multiple testing.

As suggested, we have addressed the problem and report the corrected p-value across the models. Thus obtaining a new threshold for significance of 0.0143 (0.10/7).

Table 3 & 4 should be merged into one. You should also provide the confidence interval for the K and W coefficients.

It is not possible provide confidence interval for K and W coefficients since normality can not be assumed.

Table 5 must report all values – not just ‘significant’ ones. It’s also important to include the confidence interval around the Beta estimate, as p-values alone are essentially meaningless.

We have modified the table and reported all values. Reporting Beta and its standard error is enough to quantify the effect size and its variability. P-values and confidence intervals are both functions of Beta and its standard error, thus they held both the same value in terms of inference. 

Line 165: The W values were high, but not ‘very high’ (>0.90 would be very high).

As reported in literature very high values are between 0.81 and 1

Line 170: typo – ‘avaluated’

Amended

Line 175: typo – ‘indipendently’

Amended

Line 180: repeated word “in test social attraction test”

We have removed the sentence

Line 203: What does this mean: “positive influence on “learning predisposition” indicator in pups”?

We have removed the sentence since results were modified

You say in the abstract that “In order to better standardize the test, adjustments to the scoring protocol are recommended yet I can’t find any such recommendations anywhere.

There’s one sentence on the study limitations, which isn’t really good enough.

We recommended to adjust scoring protocol and the sentence was modified

Reviewer #2: The current manuscript examines responses of 7-week old puppies to a series of behavioural tests, examining both interobserver reliability for scoring for the tests, and the effects of various puppy-related and environment-related variables on their responses. Examining early factors and their influence on puppy and later adult behaviour is an important topic. However, I have some questions about the overall objective of the current work as well as the approach that was taken. I have summarized these concerns below, and provided specific relevant comments below in the ‘general comments’ section

1. Based on the information provided in the introduction, the rationale for this particular study is unclear and needs strengthening. Why do we need to understand how different factors influence the behaviour of young puppies when previous studies have shown that performance on tasks at this age has poor predictive power for later behaviour? I recommend that the authors reframe the introduction to clearly outline what research on puppy testing has already been published and why this particular study is necessary and important – I’m assuming it is to understand sources of variability so that test performance can be improved, but this isn’t clear from what is written.

The introduction was modified according to reviewer comments.

2. It’s unclear why the authors selected to use these particular tests for testing the puppies, so the justification needs to be strengthened. Why are these areas of assessment important, and why did they select this particular test when it hasn’t been used in any previously published work? Also, the methods that were used for testing are not described in sufficient detail for proper assessment of rigour and for replication, and some of the descriptions for scoring in Table 2 are quite vague. Were these the final descriptions that were used for the observers during scoring?

We have improved the description in Table 2 and we have described in detail the methods. We have also explained the reason for selecting these particular tests.

3. The methods and results for the regression analysis are a bit confusing and this makes it difficult to properly assess the findings and conclusions for this study. Areas where more information is needed are described in more detail below. However, based on the information provided it appears that the analysis doesn’t account for clustering, which is critical when examining litters of puppies.

We reanalysed the data and included “litter” as a random effect and results were adjusted for pseudo-replication.

General Comments

Data statement: The authors have indicated that all data are present in the manuscript, but only aggregate data is presented. The raw data is not available for review.

The raw data is now available for review

The manuscript requires further editing for grammar and spelling, particularly the discussion.

Grammar and spelling were edited

L16-19: The aims for the current study are unclear. If 7 weeks of age is too young to predict later temperament, why is it important to understand factors that influence behaviour at this age? Clarification is needed

We have clarified the aim of the study which is to evaluate inter-observer reliability and identify factors that influence behavior at this age. This information could be useful for behavioural science as well as for canine breeding management

L20: replace stimuli with tests or tasks

Amended

L21-22: I wonder if the suitability of the response depends on the purpose for which the puppy has been bred? Is there a way to word this that is descriptive rather than suggesting that a particular response is best?

This could probably be true but we have deleted the genetic part of the study. The behavioural response could be suitable or not suitable, the test is performed with this aim.

Complex indicators: What is the rationale for combining particular tasks together into complex indicators. Is there some indication that responses on these tasks are related?

In many tests there are different subtests, we have considered some subtests more related to each other than others

L27-30: The results for these complex indicators are not presented in the main body of the manuscript.

We have modified the presentation of these indicators.

Introduction: The intro is a bit confusing because the authors start out discussing the potential for early testing to predict later potential, but this does not actually relate to their objective of looking at factors that influence the behaviour of puppies during early behaviour testing. The rationale for this particular study needs to be more explained more clearly.

We have modified the introduction accordingly

4-46: The following sentences aren't really necessary for the introduction: "Analyses of behaviour normally involves measuring frequency, duration and latency of specific behaviours [3]. Rater-coding is normally done on a predetermined scale that may have only 3, 5, or 7 points [4]."

We have deleted these sentences

L47: Can you clarify why increased motivation to approach might be useful for these assessments?

We have modified the sentence as follows: 

The period between 6 and 8 weeks of development may facilitate certain testing since puppies haven’t fully developed the fear response and they can be more easily handled by unknown people

L50-53: Given the objective of this study, it is important to summarize the literature on puppy assessments that have been published to date. What have people tried, what did they find and why is it inadequate and requiring further study? There is quite a large literature on this topic (both general and applied to selection for certain programs) and few studies are cited above but very little detail is provided to develop the rationale for the current study.

We have summarized the literature on puppy assessments in order to develop the rationale for the current study

L60-61: Please clarify that these tests are used in practice for assessment, and that these aren't methods that have been used and assessed in the literature. What is your rationale for using these tests rather than other tests that have been used and assessed previously in the peer-reviewed literature?

We have explained this point more clearly

L63: Missing space in "outthe"

The space was inserted

L63-65: Is it necessary to provide this information when the authors chose not to use these categories? Instead please provide further justification for the approach that you chose to go with (ie, genetic relatedness). If you do decide to keep this information in the paper, please state clearly why this approach is inappropriate for your purposes.

We have decided to not include the approach of genetic relatedness

L69-70: The objectives for the current study could use some clarification. What was your rationale for testing these specific factors, and did you have any predictions about how these factors would affect puppy behaviour before hand?

We added the rationale for testing these specific factors

L74: Add an 's' to ethics

Amended

L85: Add an 's' to breeds

Amended

L89-90: How, specifically, was the level of maternal care evaluated in the current study?

We added: presence of licking, nursing, contact, play...

L91-93: This only accounts for 84% of observations - how were the others classified?

We have corrected the mistake

L95-96: Were the puppies tested at a particular time of day?

They were tested in the late afternoon between 4.00 and 6.00 pm, we added this information

Table 1: Please expand this table to list the specific litters and number of puppies per litter to provide a better representation of the sample.

Table was expanded

L110-113: I'm having trouble understanding the methods described here - please clarify by providing further details

Further details have been provided

Table 2: The rows on the table don't line up properly so it is difficult to determine which subtests the responses align with.

The table was improved

Table 2: Fix spelling for Appearance

Amended

Table 2: Since this isn't a series of tests that has been previously published in the literature, please describe the methods in sufficient detail that they can be replicated.

We have better described the puppy response alternatives

Table 2 - Sudden Appearance Test: What does curiosity refer to specifically? And for the final category, what constituted investigation?

We have modified the sentence as follows:

Puppy moves toward the umbrella and investigates in a excited way (sniffing and wagging his tail) (300 scores)

Table 2 - Noise test: Again, what specific behavioural responses was indicative of curiosity?

We have modified the description: Puppy listens, detects sound and moves to the sound source

L126-127: Please provide basic information and a reference for interpretation of these values. There are standard cutoffs for interpretation that are commonly used in the literature.

Reference for interpretation of these values was added

L127-129: Further details are needed on the methods used for statistical modelling. Does mean score refer to the mean of all three observers, and/or were the scores converted in some manner to categories prior to ordinal logistic regression? Also, what methods, if any, were used to account for clustering with litter (e.g., was litter included as a random effect)? We can expect that litter effects are likely quite large, so realistically your sample size is reduced to the number of litters unless you can account for clustering in some way.

We reanalyzed the data “litter” as a random effect and results were adjusted for pseudo-replication. We first calculated the mean scores assigned by 3 observers for each subtests. Then the means of the subtests were averaged for computing complex indicators and inserted in the closest category (low, medium or high).

Table 3: Why were standard deviations calculated? This seems unnecessary when Kappa and Kendall's are presented.

Standard deviations were calculated to show that scores assigned by each observer were lower than scores between different behavioural responses. This is the reason that values obtained with Kendall's were high. 

L150-152: Please move this information to the M&M.

Amended

 Does this mean that scores were re-categorized as 0-100, 101-200, 201-300 prior to analysis? What was the rationale for this decision?

Scores were re-categorized as low (100), medium (200) and high (300) since variables are non-quantitative they were re-categorized arbitrarily into these 3 groups

L153-157: The figures alone do not provide sufficient information for evaluating the results from these models. Please provide the relevant values to support the comparisons that are made within the text or in Table 5.

Relevant values were provided to support the comparisons made in the table 

Table 5: What is your justification for using a significance level of 0.1? Info should be provided in M&M

We have included this information in M&M. We have selected this level because of the small sample size.

Table 5: I'm struggling a bit to interpret the information provided for differences amongst breed groups. For example, what was the overall value for genetic group and which referent are these in comparison to? Or was each group looked at separately in comparison to others not in that group? Apologies if I'm misinterpreting how this was done but it isn't clear to me.

We have decided to not consider breed groups. 

Table 5: Sudden appearance test – typo

Amended

L164: The agreement between observers suggests reasonable agreement for scoring, but I'm not sure how this relates to clear representations of puppy behaviour. Please explain

The agreement between observers implies an observation of a similar behaviour. If test are not performed to discriminate behaviours, observers would be not able to score the correct behaviour in a reliable way.

L165: What are your interpretations of these statistics based on - please provide references.

Reference were provided 

L170: While the Kendall's W values are good, the Kappa values are in the moderate agreement range. It might be worth discussing why both were used, and differences in interpretation between the two.

We have modified the discussion according this comment, Kappa is about perfect agreement, either judges assign the same score or they do not agree with each other. W instead also takes into account by how much the scores diverge to express the agreement.

L189: Can you be more specific about what aspects of behaviour were affected in this previous study?

Amended

Discussion: The discussion might benefit from additional consideration of how the current puppy tests are similar/dissimilar from previous research in this area.

We have modified the discussion accordingly.

L215-216: The final sentence of the conclusion is not supported by the data presented.

We have modified the final sentence

L268: This reference is not available at the link provided.

We have replaced reference

Figure 6: Should the blue label read Unsuitable or Unstimulating?

We have deleted the figure since the results have been modified by further statistical analysis

---

## [Decision Letter · Decision Letter 1]

17 Jun 2020

PONE-D-20-02339R1

Behavior test for seven-week old puppies ( Canis familiaris ): inter-rater reliability and factors associated with test performance

PLOS ONE

Dear Dr. Alberghina

Thank you for submitting your manuscript to PLOS ONE. A few minor revisions have been suggested by the reviewers. Therefore, we invite you to submit a revised version of the manuscript that addresses the points raised during the review process.

Many thanks for resubmitting your manuscript to PLOS One

The manuscript has been reviewed and the reviewers have requested some further minor revisions

If you could make the minor revisions, and write a response to reviewers, then your manuscript can be reviewed rapidly upon re-submission

I wish you the best of luck with your revisions

Hope you are keeping safe and well in these difficult times

Thanks

Simon

We look forward to receiving your revised manuscript.

Kind regards,

Simon Clegg, PhD

Academic Editor

PLOS ONE

Reviewers' comments:

Reviewer's Responses to Questions

**Comments to the Author**

1. If the authors have adequately addressed your comments raised in a previous round of review and you feel that this manuscript is now acceptable for publication, you may indicate that here to bypass the “Comments to the Author” section, enter your conflict of interest statement in the “Confidential to Editor” section, and submit your "Accept" recommendation.

Reviewer #1: (No Response)

Reviewer #3: All comments have been addressed

2. Is the manuscript technically sound, and do the data support the conclusions?

Reviewer #1: Yes

Reviewer #3: Yes

3. Has the statistical analysis been performed appropriately and rigorously? 

Reviewer #1: Yes

Reviewer #3: Yes

4. Have the authors made all data underlying the findings in their manuscript fully available?

Reviewer #1: Yes

Reviewer #3: Yes

5. Is the manuscript presented in an intelligible fashion and written in standard English?

Reviewer #1: Yes

Reviewer #3: Yes

6. Review Comments to the Author

Reviewer #1: This is the 2nd round of reviewing for this manuscript. The authors have done well to address extensive comments from both reviewers, which have included amending the manuscript title and re-doing their statistical analysis. Much of the text has been re-written or amended so I have had to read it again with fresh eyes. Specific comments for minor improvements are given below.

Abstract:

The age of the dogs should be stated in the abstract (I know it’s in the title, but it still needs to be in the abstract).

Perhaps the term ‘aggregate indicators’ is a more precise description than ‘complex indicators’?

The abstract is much improved, well done to the authors.

The Introduction is much improved and the aims/purpose of the study much clearer.

Methods, some minor comments:

Whilst the experimenter and camera person may have been male, please consider using gender neutral terminology to describe the protocol i.e. ‘A second person filmed the test for subsequent video analysis’ (first paragraph under Procedure), and in Table 2 change ‘him’ to ‘them’ (in the response to Following)

The way maternal care was captured has not been described still. What exactly did breeders score? This needs to be described if anyone were to replicate (or improve) upon the way it was judged.

Whilst I do understand what you mean now when you say this, I think it can still be made clearer for a new reader: ‘The interval between behavioural responses for each test was obtained dividing by 5, 4 or 3 depending of the number of observed reactions.’

Perhaps this wording will help: As the maximum score was 300, the score assigned to the behavioural responses for each test was obtained by dividing 300 by 5, 4 or 3 depending of the number of response categories for the test.

Results:

Here: ‘Litter size significantly influenced the social attraction (P=0.03), following and sudden appearance (P=0.06)’ it would be good to clarify that the p value was the same for Following and Sudden appearance as it looks like the value simply wasn’t stated for Following.

P=0.06 is being stated as ‘significant’ and in the tables there are (slightly hard to find) notes to say alpha = 0.10 but this isn’t stated anywhere in the text with justification for this. I know that this was done due to the small sample size from the replies to reviewers comments, but new readers also need to know this.

Discussion:

Whilst these statements are true, they are just hanging there as their own isolated paragraph and aren’t connected to anything else: “Dogs raised in domestic environments were less likely to develop fear and aggression towards unfamiliar people compared to dogs raised in non-domestic environments [24]. Sufficient exposure to relevant stimuli during the early socialisation period appears to be associated with lower fearfulness and aggression in dogs [25].”

Conclusion: I am not convinced that you can recommend the test for use based upon the results of this study alone as you do here “This simple test can be recommended for all puppies.”. This study shows that there is sufficient inter-rater reliability in the test, and that it captures some variance that is associated with other factors, but what are you recommending it for use for? I think the most you can conclude is that you have successfully designed a score system for the test that can be used reliably by different people and for quantitative analysis.

Reviewer #3: I was invited as a second reviewer, presumably due to the unavailability of the original reviewer (?). I thoroughly enjoyed reading the manuscript, and found it very interesting. It appears that the intense first review comments have mostly been addressed. This is a very interesting paper, and I look forward to seeing it published. Most of my comments are only minor.

This appears to be a major development in the behavioural assessment field, and one which I was privileged to read. I praise the authors on an excellent study, and a good manuscript, and offer my thanks for this study, and offer my best wishes for the future, and for your safety in the coronavirus pandemic.

You have a really nice abstract

Introduction

The introduction is nice, clear and well written. It has a nice flow and clearly states the aims of the study. Perhaps a little bit of extra detail on the modification maybe useful, but I do not feel strongly about that, and I will leave that up to your judgement.

The period between 6 and 7 weeks of development may facilitate certain testing, since puppies haven’t fully developed the fear imprinting response and they can be more easily handled by unknown people [3]. Please add in the comma between testing and since.

Methods

Again, generally well written, but a few minor points

I feel that the methods would be better written in a neutral gender. It almost sounds like the only way it would work is with a male.

You mention the maternal experience and the environment. How was this analysed? Was this by yourselves, or the breeder, and if it is the latter, how can you be sure that it was honest or standardised?

I particularly struggled to follow the following ‘The interval between behavioural responses for each test was obtained dividing by 5, 4 or 3 depending of the number of observed reactions’. Is it possible to reword it please?

Table 1- could you group these by size so it is easier to read?

Results

Some of the tables would be nicer as a proper table rather than as it is – i.e. with lines like Table 1

Variance due to the litter was very low in all tasks, except the sudden appearance and noise subtests. Add in the comma between tasks and except

You have a P value of 0.06 being statistically significant. It is more common to use 0.05. Is there a reason why this value was chosen? I feel that this should be stated somewhere in the text, probably the methods (unless I missed it)

Discussion

Again, I really like the discussion

There are a few random statements which just appear dumped in certain places which could be better integrated elsewhere in the discussion

I think that your conclusion is good, but maybe over egging it somewhat. Perhaps consider toning it down.

7. PLOS authors have the option to publish the peer review history of their article (what does this mean?). If published, this will include your full peer review and any attached files.

Reviewer #1: No

Reviewer #3: No

---

## [Author Response · Author response to Decision Letter 1]

1 Jul 2020

29/06/2020 Messina,

Dear Editor and Reviewers,

Thank you very much for reviewing our manuscript titled “Behavior test for seven-week old puppies (Canis familiaris): inter-rater reliability and factors associated with test performance”.

 We are happy to know that reviewers found the manuscript improved. We are including all useful reviewers’ suggestions and clarifying the text when needed. We are confident that the new version of the manuscript will be greatly improved.

Please, find below the referees’ comments (black font) and our responses (blue font) inserted after each comment. 

Looking forward hearing from you soon. 

Best regards,

Daniela Alberghina

Reviewers' comments

Abstract:

The age of the dogs should be stated in the abstract (I know it’s in the title, but it still needs to be in the abstract).

Perhaps the term ‘aggregate indicators’ is a more precise description than ‘complex indicators’?

The age of the dogs was stated and we agree with reviewer that the term “aggregate” is more appropriate than “complex”

The abstract is much improved, well done to the authors.

Many thanks

The Introduction is much improved and the aims/purpose of the study much clearer.

Methods, some minor comments:

Whilst the experimenter and camera person may have been male, please consider using gender neutral terminology to describe the protocol i.e. ‘A second person filmed the test for subsequent video analysis’ (first paragraph under Procedure), and in Table 2 change ‘him’ to ‘them’ (in the response to Following)

Amended

The way maternal care was captured has not been described still. What exactly did breeders score? This needs to be described if anyone were to replicate (or improve) upon the way it was judged.

We have asked to the breeder the presence of adequate maternal behaviour (e.g. mother-pup interaction during feeding sessions, licking, contact, play, movement towards and away from the puppy) and their response was classified as adequate or inadequate maternal behaviour

Whilst I do understand what you mean now when you say this, I think it can still be made clearer for a new reader: ‘The interval between behavioural responses for each test was obtained dividing by 5, 4 or 3 depending of the number of observed reactions.’

Perhaps this wording will help: As the maximum score was 300, the score assigned to the behavioural responses for each test was obtained by dividing 300 by 5, 4 or 3 depending of the number of response categories for the test.

Thank you for suggestion!

Results:

Here: ‘Litter size significantly influenced the social attraction (P=0.03), following and sudden appearance (P=0.06)’ it would be good to clarify that the p value was the same for Following and Sudden appearance as it looks like the value simply wasn’t stated for Following.

P=0.06 is being stated as ‘significant’ and in the tables there are (slightly hard to find) notes to say alpha = 0.10 but this isn’t stated anywhere in the text with justification for this. I know that this was done due to the small sample size from the replies to reviewers comments, but new readers also need to know this.

We have added this information in the text: Due to the small sample size P values <0.10 were considered significant. 

Discussion:

Whilst these statements are true, they are just hanging there as their own isolated paragraph and aren’t connected to anything else: “Dogs raised in domestic environments were less likely to develop fear and aggression towards unfamiliar people compared to dogs raised in non-domestic environments [24]. Sufficient exposure to relevant stimuli during the early socialisation period appears to be associated with lower fearfulness and aggression in dogs [25].”

We have added an additional sentence to link these two statements to our findings. 

Conclusion: I am not convinced that you can recommend the test for use based upon the results of this study alone as you do here “This simple test can be recommended for all puppies.” This study shows that there is sufficient inter-rater reliability in the test, and that it captures some variance that is associated with other factors, but what are you recommending it for use for? I think the most you can conclude is that you have successfully designed a score system for the test that can be used reliably by different people and for quantitative analysis.

Thank you for suggestion! We have modified conclusions accordingly

Reviewer #3: I was invited as a second reviewer, presumably due to the unavailability of the original reviewer (?). I thoroughly enjoyed reading the manuscript, and found it very interesting. It appears that the intense first review comments have mostly been addressed. This is a very interesting paper, and I look forward to seeing it published. Most of my comments are only minor.

 Thank you for your kind comments

This appears to be a major development in the behavioural assessment field, and one which I was privileged to read. I praise the authors on an excellent study, and a good manuscript, and offer my thanks for this study, and offer my best wishes for the future, and for your safety in the coronavirus pandemic.

We thank you and wish the same with your safety

You have a really nice abstract

:)

Introduction

The introduction is nice, clear and well written. It has a nice flow and clearly states the aims of the study. Perhaps a little bit of extra detail on the modification maybe useful, but I do not feel strongly about that, and I will leave that up to your judgement.

We have believed that modification are enough described but please give us other suggestion about that.

The period between 6 and 7 weeks of development may facilitate certain testing, since puppies haven’t fully developed the fear imprinting response and they can be more easily handled by unknown people [3]. Please add in the comma between testing and since.

Amended

Methods

Again, generally well written, but a few minor points

I feel that the methods would be better written in a neutral gender. It almost sounds like the only way it would work is with a male.

We have rewritten in neutral gender

You mention the maternal experience and the environment. How was this analysed? Was this by yourselves, or the breeder, and if it is the latter, how can you be sure that it was honest or standardised?

We can not be sure for information about adequate maternal behaviour. For this reason the original sentence in the conclusion session was: In this study, our results did not show an effect of adequate maternal behaviour on performance response to subtests. This could be due to the subjectivizes of the breeders asked to judge this behaviour. 

We have added the following sentence: For future studies, maternal behaviour should also be recorded directly to determine whether it has an impact on responses to these test. More specifically, in order to have a better standardization of observed behaviours, maternal behavior should be recorded 1 day per week continuously every second hour over a 24-hour period during the first 3weeks postpartum as described by Foyer et al. [20] 

I particularly struggled to follow the following ‘The interval between behavioural responses for each test was obtained dividing by 5, 4 or 3 depending of the number of observed reactions’. Is it possible to reword it please?

Amended

Table 1- could you group these by size so it is easier to read?

Amended

Results

Some of the tables would be nicer as a proper table rather than as it is – i.e. with lines like Table 1

Amended

Variance due to the litter was very low in all tasks, except the sudden appearance and noise subtests. Add in the comma between tasks and except

Amended

You have a P value of 0.06 being statistically significant. It is more common to use 0.05. Is there a reason why this value was chosen? I feel that this should be stated somewhere in the text, probably the methods (unless I missed it)

We have added this information in the text

Discussion

Again, I really like the discussion

There are a few random statements which just appear dumped in certain places which could be better integrated elsewhere in the discussion

I think that your conclusion is good, but maybe over egging it somewhat. Perhaps consider toning it down 

We have modified the conclusion accordingly your suggestion

---

## [Editor Report · Decision Letter 2]

6 Jul 2020

Behavior test for seven-week old puppies ( Canis familiaris ): inter-rater reliability and factors associated with test performance

PONE-D-20-02339R2

Dear Dr. Alberghina

We’re pleased to inform you that your manuscript has been judged scientifically suitable for publication and will be formally accepted for publication once it meets all outstanding technical requirements.

Kind regards,

Simon Clegg, PhD

Academic Editor

PLOS ONE

Additional Editor Comments:

Many thanks for submitting your manuscript to PLOS One

I have read through the manuscript, and as you have addressed all comments and the manuscript reads well, I have recommended your manuscript for publication

You should hear from the Editorial Office

It was a pleasure working with you and I wish you all the best for your future research

Hope you are keeping safe and well in these difficult times

Thanks

Simon

---

## [Editor Report · Acceptance letter]

8 Jul 2020

PONE-D-20-02339R2 

Behavior test for seven-week old puppies ( Canis familiaris ): inter-rater reliability and factors associated with test performance 

Dear Dr. Alberghina:

I'm pleased to inform you that your manuscript has been deemed suitable for publication in PLOS ONE. Congratulations! Your manuscript is now with our production department. 

Kind regards, 

on behalf of

Dr. Simon Clegg 

Academic Editor

PLOS ONE